# Controlled Attenuation Parameter Is Associated with a Distinct Systemic Inflammatory Milieu after Clearance of HCV Infection

**DOI:** 10.3390/biomedicines11061529

**Published:** 2023-05-25

**Authors:** Yanqin Du, Tanvi Khera, Zhaoli Liu, Magdalena Tudrujek-Zdunek, Anna Dworzanska, Markus Cornberg, Cheng-Jian Xu, Krzysztof Tomasiewicz, Heiner Wedemeyer

**Affiliations:** 1Department of Gastroenterology and Hepatology, University Hospital Essen, University Duisburg-Essen, 45147 Essen, Germany; yanqin.du@163.com (Y.D.); tanvikhera@gmail.com (T.K.); 2Department of Infectious Diseases, Union Hospital, Tongji Medical School, Huazhong University of Science and Technology, Wuhan 430022, China; 3Department of Gastroenterology, Hepatology, Infectious Diseases and Endocrinology, Hannover Medical School, 30625 Hannover, Germany; cornberg.markus@mh-hannover.de (M.C.); xu.chengjian@mh-hannover.de (C.-J.X.); 4International AIDS Vaccine Initiative (IAVI), 122002 Gurugram, Haryana, India; 5Centre for Individualized Infection Medicine (CiiM), a Joint Venture between the Helmholtz Centre for Infection Research (HZI) and Hannover Medical School (MHH), 30625 Hannover, Germany; zhaoli.liu@helmholtz-hzi.de; 6TWINCORE Centre for Experimental and Clinical Infection Research, a Joint Venture between the Helmholtz Centre for Infection Research (HZI) and the Hannover Medical School (MHH), 30625 Hannover, Germany; 7Department of Infectious Diseases, Medical University of Lublin, 20-081 Lublin, Poland; magdalena.tudrujek@gmail.com (M.T.-Z.); annadw8@gmail.com (A.D.); tomaskdr@poczta.fm (K.T.); 8Department of Internal Medicine and Radboud Center for Infectious Diseases, Radboud University Medical Center, 6525 GA Nijmegen, The Netherlands; 9Excellence Cluster Resist, Hannover Medical School, 30625 Hannover, Germany

**Keywords:** steatosis, chronic hepatitis C virus, sustained viral response, soluble inflammatory mediators

## Abstract

Hepatitis C virus (HCV) infection is closely associated with lipid metabolism defects along with a high prevalence of hepatic steatosis. After HCV clearance, steatosis persists in many patients. However, the reasons behind this phenomenon are not completely clear. To investigate the association between 92 soluble inflammatory mediators (SIMs) and the steatosis grade, we made use of a cohort of 94 patients with chronic HCV infection who cleared HCV after direct-acting antiviral agent (DAA) treatment. Patients were classified into three groups according to their controlled attenuation parameter (CAP). CAP is associated with ALT, γ-GT and liver stiffness after HCV clearance. While stem cell factor (SCF) and tumor necrosis factor ligand superfamily member 12 (TWEAK) levels were significantly reduced in patients with CAP > 299 dB/m, the levels of fibroblast growth factor (FGF)-21 and interleukin-18 receptor 1 (IL-18R1) were higher in those patients at week 96 after virus clearance. These four markers also showed a linear correlation with CAP values. FGF-21 levels correlated with CAP only after HCV clearance. Taken together, these four biomarkers, namely SCF, TWEAK, FGF-21 and IL-18R1, are associated with CAP status after virus clearance. A potential role of these proteins in the pathogenesis of post-sustained viral response (SVR) nonalcoholic steatohepatitis requires further investigation.

## 1. Introduction

Fatty liver disease is the most common chronic liver disease worldwide, which includes alcoholic fatty liver disease (AFLD) and nonalcoholic fatty liver disease (NAFLD), with the prevalence of 6% and 23.7% in Europe [1], respectively. NAFLD is driven by metabolism syndrome, associated with obesity, insulin resistance, and hyperlipidemia [2], and is projected to increase by up to 56% between 2016 and 2030 in several European countries [3]. NAFLD is a systemic inflammatory disease, and it can develop from fat accumulation (simple steatosis) to steatohepatitis and even advanced diseases, such as cirrhosis and hepatocellular carcinoma (HCC).

Hepatitis C virus (HCV) is a hepatotropic virus, causing both acute and chronic infection in human beings. It is estimated that 55–85% of individuals with HCV infection will develop into a persistent infection [4]. HCV infection is closely associated with lipid metabolism defects throughout the viral lifecycle [5]. The prevalence of hepatic steatosis in chronic HCV infection ranges from 50 to 80%, 2.5-fold higher than in the general population and in the other forms of chronic liver disease [6]. Two main types of steatoses in patients with HCV infection have been defined: the metabolic type of steatosis associated with a high body mass index (BMI), hyperlipidemia and insulin resistance; and the virally induced steatosis that is linked with HCV genotype 3 infection [7]. Over the past few years, numerous direct-acting antiviral (DAA) agents have been used in patients with HCV infection with high efficacy and safety; however, the steatosis disorder remains in patients with HCV after successful virus clearance, especially for the metabolic type of steatosis [8,9]. Moreover, obesity, diabetes, liver cirrhosis, alcohol consumption, and endpoints achieving sustained viral response (SVR) are independently associated with persistently increased liver enzymes after HCV clearance [9]. Studies that compare the non-invasive method of measuring hepatic steatosis, controlled attenuation parameter (CAP), to the golden standard, liver biopsy, have illustrated that CAP has high diagnostic accuracy for detecting hepatic steatosis [10,11,12,13]; however, studies evaluating steatosis and fibrosis in liver biopsies after SVR are still lacking [14].

HCV infections are often accompanied with alteration in the hepatic and systemic inflammatory state, characterized by the upregulation of several soluble mediators. Several studies have investigated the inflammatory cytokines and chemokine milieu in both acute [15,16] and chronic HCV infection [17,18]. Acute and chronic HCV infection significantly disrupts the milieu of soluble inflammatory mediators (SIMs) [18,19]; however, DAA-induced HCV clearance is not able to completely restore the altered SIMs [18,19]. Interestingly, it has also been reported that DAA treatment is followed by a substantial weight gain in one-third of patients during long-term follow-up [20]. Previous data have shown that some soluble mediators can regulate the development of steatosis, such as soluble CD163 [21], monocyte chemoattractant protein (MCP)-1, interleukin (IL)-8, and IL-6 [22]. Therefore, it is of potential interest to investigate whether SIMs are associated with the steatosis status in patients with chronic HCV infection after virus clearance.

In the present study, we made use of a well-characterized cohort of patients with chronic HCV infection who were cured with DAA. By comparing the levels of SIMs in patients with different grades of steatosis at long-term follow up (96 w) after viral clearance, we may identify markers as indicators for steatosis.

## 2. Materials and Methods

### 2.1. Patient Materials

In this study, 94 patients with chronic hepatitis C were included. All patients were genotype 1b and were negative for human immunodeficiency virus (HIV) and hepatitis B virus (HBV). These patients were monitored before, during, and after DAA treatment in the outpatient clinic of the Department of Infectious Diseases at the Medical University of Lublin (Lublin, Poland). Patients were treated either with a three-drug regimen containing a protease inhibitor against HCV (ombitasvir (OBV) + paritaprevir/ritonavir (PTV/r) and dasabuvir (DSV)) or a protease-free regimen (sofosbuvir (SOF) and ledipasvir (LDV)) with or without ribavirin (RBV) for 8, 12 or 24 weeks. Peripheral blood samples were collected at baseline (before the start of the therapy), end of treatment (EOT), 36 weeks, and 96 weeks after the start of treatment (Figure 1A). All the patients achieved a sustained viral response (SVR) after treatment. Blood plasma was collected from ethylenediaminetetraacetic acid (EDTA)-treated peripheral blood samples and stored at −80 °C for later analysis. Samples were collected from patients from 1 January 2016 to 31 October 2018.

Previous studies have demonstrated that CAP has high diagnostic accuracy for detecting hepatic steatosis [10,11,12,13]. Therefore, patients were examined by Transient elastography (FibroScan) using Fibroscan^TM^ 502 Echosens, which includes vibration-controlled transient elastography to assess liver stiffness measurement and controlled attenuation parameter (CAP). Patients were classified into three groups, as previously reported [12], based on the values of CAP at follow-up on week 96: CAP < 250 dB/m; CAP 250–299 dB/m; CAP > 299 dB/m. All patients gave informed written consent for the study. The protocols for sample collection and investigation were reviewed and proved by the local ethics committee of the Medical University of Lublin (KE-0254/311/2018).

### 2.2. Protein Quantification by Proximity Extension Analysis

Plasma samples were thawed, and 20 μL plasma of each sample was seeded into 96-well plate and were sent to O-link AB for Proseek inflammation panel analysis. The proximity extension analysis (PEA) simultaneously measures 92 human inflammation-related proteins as previously described [19]. Briefly, a pair of oligonucleotide-labeled antibodies (Proseek probes) binds to the target protein in the plasma sample. When the two Proseek probes are in proximity, a new polymerase chain reaction target sequence is formed by a proximity-dependent DNA polymerization event. This complex is subsequently detected and quantified using a standard real-time polymerase chain reaction. The generated normalized protein expression unit is on a log2 scale, where a larger number represents a higher protein level in the sample. The Cq values from a DNA extension control are subtracted from the measurement of Cq value, an interpolate control is corrected for and finally a correction factor is subtracted to yield a normalized protein expression (NPX) value, which is log2 transformed. More details about the limit of detection, reproducibility, and validation are available at the Olink Proteomics website (http://www.olink.com/products-services/target/inflammation/ (accessed on 1 January 2019)).

### 2.3. Statistical Analysis

Data were analyzed using GraphPad Prism v8 (Graph Pad Software, La Jolla, CA, USA). All data were evaluated for their statistical distribution by using the Shapiro–Wilk normality test or the D’Agostino-Pearson omnibus normality test. Statistical significance between two groups was assessed by parametric Student’s t-test for normally distributed values, or by nonparametric Wilcoxon test or Mann–Whitney test for values that did not show a normal distribution. Statistical significance among multiple groups was assessed by parametric ANOVA and multiple t test for normally distributed values or by nonparametric Kruskal–Wallis test and Dunn’s multiple comparison test for values that did not show a normal distribution. The heatmap (pheatmap package) and volcano plots (ggplot2 package) were made using R language. The statistical test used for each analysis is mentioned in the respective figure legends. A *p* value less than 0.05 was considered statistically significant. * *p* < 0.05; ** *p* < 0.01; *** *p* < 0.001.

## 3. Results

### 3.1. Hepatic Steatosis Remains in Some of Patients with Chronic HCV Infection despite Successful SVR

A previous study demonstrated that chronic HCV patients with genotype 1 or 2 infection had decreased liver steatosis after achieving SVR [23]. Therefore, we first analyzed the status of hepatic steatosis of patients of chronic HCV infection at 96 w after the initiation of DAA treatment. We classified the patients into three groups based on their CAP values at 96 w (Figure 1B) and compared the clinical indicators at baseline and 96 w among the three groups. The clinical characteristics are presented in Table 1. Of note, the CAP value in 41.4% of patients (39 out of 94) increased more than 20 dB/m at 96 w compared to that at baseline (Figure 1B). In line with this, we also found elevated total cholesterol, triglycerides, and LDL in patients at 96 w compared to those at baseline (Appendix A). Interestingly, the BMI values, liver stiffness, alanine aminotransferase (ALT), glutamyl transpeptidase (GGT), and glucose were significantly higher in patients of CAP > 300 dB/m than those in patients with CAP 250–299 dB/m or patients with CAP < 250 dB/m (Figure 1C). Consistently, CAP values significantly correlated with BMI values, liver stiffness (Fibroscan values), ALT, aspartate aminotransferase (AST), and GGT values (Figure 1D). Taken together, high CAP values induced by liver inflammation and metabolic syndrome remained in some of the HCV patients even after the virus clearance.

### 3.2. Four SIMs Differed in Patients with Different Steatosis Status after Successful HCV Clearance

To investigate the expression of SIMs, we made use of proximity extension analysis as previously described [19]; we curated the data by eliminating measured proteins and samples that did not pass the quality controls. We retained 72 markers for downstream analysis from a panel of 92 and categorized them based on their function, chemokines (*n* = 19), interleukins (*n* = 12), TNF-associated cytokines (*n* = 6), growth factor (*n* = 13), ligands (*n* = 9) and others (*n* = 13) (Figure 1A, Table 2).

In the next step, we explored the SIM expression in patients with different steatosis grade at 96 w. This is depicted by a heatmap that shows 72 SIMs of each patient based on the normalized protein expression values (NPX) (Figure 2A). We did not observe a cluster pattern of steatosis status with age, sex, cirrhosis and protein expression value. Next, we identified differential protein analysis by linear regression after correcting age, gender and BMI. As shown in volcano plots (Figure 2B), compared to the group of CAP < 250 dB/m, both patients with CAP 250–299 dB/m and patients with CAP > 299 dB/m expressed lower levels of tumor necrosis factor ligand superfamily member 12 (TWEAK). Compared to both patients with CAP < 250 dB/m and patients with CAP 250–299 dB/m, patients with CAP > 299 dB/m exhibited higher levels of IL-18R1 and lower levels of stem cell factor (SCF) (Figure 2B). Moreover, a distinct clustering of three patient groups could be seen in the principal component analysis (PCA) plot based on all significantly different values (*p* ≦ 0.05) when using the comparison among multiple groups (Appendix A). Of note, four parameters, comprising two growth factors (SCF, fibroblast growth factor (FGF)-21), one interleukin (IL-18R) and one TNF-associated cytokine (TWEAK) were significantly different among the patient groups, based on PCA (Figure 2C). The expression of SCF and TWEAK were significantly lower in patients with CAP > 250 dB/m compared to patients with CAP < 250 dB/m (Figure 2C). Conversely, the expression of FGF-21 and IL-18R1 was higher in patients with CAP > 250 dB/m compared to those in patients with CAP < 250 dB/m (Figure 2C).

To confirm whether these four markers are associated with the status of steatosis, we performed linear regression between these markers and CAP values. Consistently, the levels of SCF and TWEAK negatively correlated with the CAP values and the levels of FGF-21 and IL-18R1 positively correlated with CAP values (Figure 3). Therefore, these four biomarkers, SCF, TWEAK, FGF-21 and IL-18R1, might specifically be associated with high CAP values after virus clearance.

### 3.3. SCF, TWEAK, FGF-21, and IL-18R1 Were Specifically Related to Steatosis at 96 w after Virus Clearance

To determine whether these four markers already showed difference among different steatosis groups before 96 w, we analyzed the kinetics of these four markers. Among these four markers, the levels of SCF and TWEAK were significantly lower in patients with CAP 250–299 dB/m and patients with CAP > 299 dB/m than those in patients with CAP < 250 dB/m at baseline, whereas the expression of IL-18 R1 was significantly higher in patients with CAP > 299 dB/m compared to patients with CAP < 250 or 250–299 dB/m at baseline; however, no significant difference was observed regarding the expression of FGF among the three groups at baseline (Figure 4). Interestingly, SCF, TWEAK, and IL-18 R1 that differed among the three groups at baseline all exhibited similar levels among the three groups at the end of treatment and 36 w, which may suggest that their expression is also related to the virus activity. Of note, all there four markers displayed significant difference again among the three groups at 96 w (Figure 4).

A previous study demonstrated that BMI was an important confounding factor when evaluating steatosis with CAP [24]. Therefore, we also compared the kinetics of these four markers in patients with a different pattern of BMI change. As shown in Appendix A, although TWEAK and FGF-21 displayed different levels among the three groups at the end of treatment, no significant difference was observed regarding these four markers among the three groups both at baseline and 96 w.

## 4. Discussion

In this study, we comprehensively analyzed the expression of SIMs in patients with chronic HCV infection in relation to the status of CAP values at the long-term follow up after the virus clearance. Overall, our data suggested that four markers, SCF, TWEAK, FGF-21, and IL-18R1 differed in the different status of CAP value at 96 w after virus clearance, and these four markers were all significantly correlated with the CAP value.

A previous study demonstrated that hepatic steatosis is common in Egyptian HCV patients, and it increases after HCV eradication with DAAs [25]. In line with this, our study also illustrated that steatosis disorder remained in most patients despite successful HCV clearance. In fact, the metabolic type of steatosis in HCV patients, except for genotype 3, is mainly linked to factors such as a high BMI, hyperlipidemia, and insulin resistance [26,27], similar to HCV-uninfected individuals. Moreover, an unexpected elevation in hepatic steatosis after DAA treatment has been found to be linked with lipid storm, elevation of LDL, and sdLDL [28]. Consistently, our study also found elevated total cholesterol, triglycerides, and LDL in patients at 96 w compared to those at baseline (Appendix A), which may have contributed to the increased hepatic steatosis; however, the reasons for the increased total cholesterol, triglycerides, and LDL after HCV eradication still need to be elucidated.

At 96 w, we observed four markers that differed among different status of CAP values. TWEAK has been reported to reduce lipid accumulation in human liver cells, and the decreased soluble TWEAK concentrations are independently associated with the presence of NAFLD [29]. However, the overexpression of FGF-21 ameliorates obesity and liver steatosis [30]. Consistently, in our study, the expression of TWEAK was lower in patients with CAP > 299 dB/m, while the expression of FGF-21 was higher in patients with CAP > 299 dB/m. Soluble IL-18 receptor 1 has been reported as a biomarker in the diagnosis of rheumatoid arthritis [31]. Its ligand, IL-18, has been associated with hepatic steatosis and elevated liver enzymes in people with HIV infection [32] and has been used as a predictive marker in liver steatosis in obese children [33]. Although the expression of IL-18 did not show significant difference among the three groups in our cohort, the elevated IL-18R1 may promote steatosis development through increased interaction with IL-18 in patients with CAP > 299 dB/m. SCF has been reported to regulate hemopoietic stem cells homing and proliferation [34]. Besides that, SCF decreased in the liver injury, suggesting its proliferative effects on hepatocytes [35]. In our study, we found significantly lower expression of SCF in patients with CAP > 299 dB/m, which might be associated with the impaired liver regeneration in these patients [36].

Previously, our group also found that soluble inflammatory milieu was not restored even in the long-term follow up after HCV clearance upon DAA treatment with or without ribavirin [18,19]. This observation may explain some level of persistence of metabolic comorbidities despite viral clearance in our study. The levels of SCF and TWEAK negatively correlated with CAP value, while FGF-21 and IL-18R1 positively correlated with the CAP value at 96 w after HCV clearance (Figure 3). This is in line with previously published studies that showed FGF-21 was increased in obesity and NAFLD [37] and that decreased soluble TWEAK levels are associated with increased risk of diabetes [38,39] and metabolic syndrome [40]. In our study, we also found that SCF, TWEAK, and IL-18R were also associated with steatosis before treatment, while FGF-21 was not. It suggests that FGF-21 may have a specific role in NASH, while the difference of SCF, TWEAK, and IL-18R at 96 w may be partially influenced by previously infected HCV. Furthermore, we previously observed the upregulation of IL-18 in NASH patients compared to healthy controls [18], which may be associated with the increased expression of IL-18R1 at 96 w in our study. However, the expression of SCF was not significantly decreased in NASH patients [18]. This is not consistent with the significant downregulation of SCF in patients with CAP > 299 dB/min our study. This may be due to different number of patients in the two cohort. The previous cohort contains only 5 healthy controls and 20 NASH patients, while our cohort included 94 patients. Thus, detailed effects of SCF on NASH, obesity, and metabolic syndrome need to be confirmed.

This study has several strengths and limitations as discussed below. Firstly, the study includes a well-characterized and homogenous cohort of patients, with only HCV-cleared patients included. Secondly, a large panel of cytokines and chemokines were analyzed with a very sensitive technique. On the other hand, few limitations should also be considered. Firstly, we only use CAP value for evaluating the steatosis status, which is not based on the golden standard, histology. In addition, BMI may have contributed to different CAP results as previously reported [24]. Therefore, the findings in our study requires validation in future research. The second limitation is the lack of the mechanistic studies investigating the role of the four biomarkers in the progression of steatosis. Thirdly, only genotype 1b was included in this study. Therefore, future studies should take other genotypes into consideration.

In conclusion, a high CAP values remains in most of the patients despite successful viral clearance. Importantly, four biomarkers, namely SCF, TWEAK, FGF-21, and IL-18R1, may specifically be associated with CAP status after virus clearance.

## Figures and Tables

**Figure 1 biomedicines-11-01529-f001:**
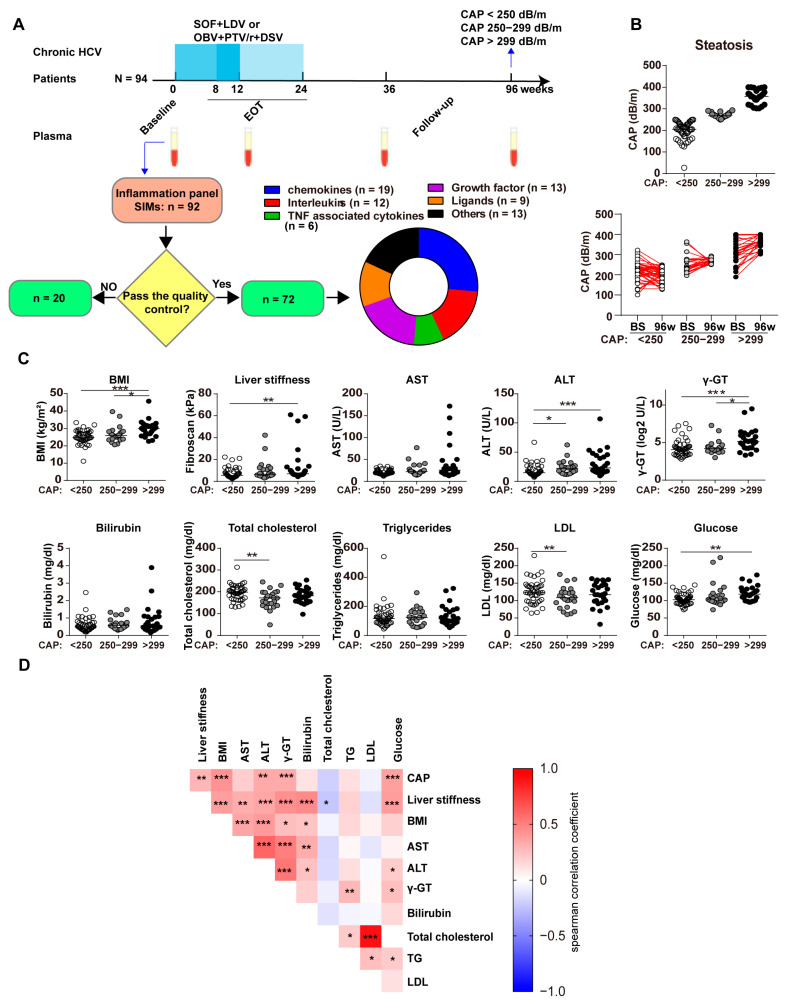
Study design and patient characteristics. (**A**) Patients with chronic HCV infection (*n* = 94) were treated with ledipasvir plus sofosbuvir or omibitasvir with paritaprevir/ritonvavir for 8 or 12 or even 24 weeks. Plasma was collected from all these patients at baseline, end of treatment, week 36, and week 96. Proximity extension assay was performed profiling a total of 92 soluble inflammatory molecules, of which 72 markers were selected and categorized to different groups. The remaining 20 SIMs were below the limit of detection or did not pass the internal quality control and were excluded from analysis. (**B**) Patients were classified into three groups according to their steatosis status based on CAP values at 96 w: CAP < 250 dB/m (*n* = 44), CAP 250–299 dB/m (*n* = 23), CAP > 299 dB/m (*n* = 27). The CAP values in different groups at 96 w were summarized. (**C**) Clinical parameters of these patients at 96 w were summarized. (**D**) Correlation matrix showed the correlation between different clinical parameters at 96 w. Statistical analysis was performed via ANOVA with Kruskal–Wallis test followed by Dunn’s multiple comparison test. * *p* < 0.05, ** *p* < 0.01, *** *p* < 0.001. The horizontal bars represent the median. Abbreviations: BS, baseline; BMI, body mass index; AST, aspirate aminotransferase; ALT, alanine aminotransferase; γ-GT, gamma-glutamyl transferase; SIM, soluble inflammatory mediator; LDL, low-density lipoprotein.

**Figure 2 biomedicines-11-01529-f002:**
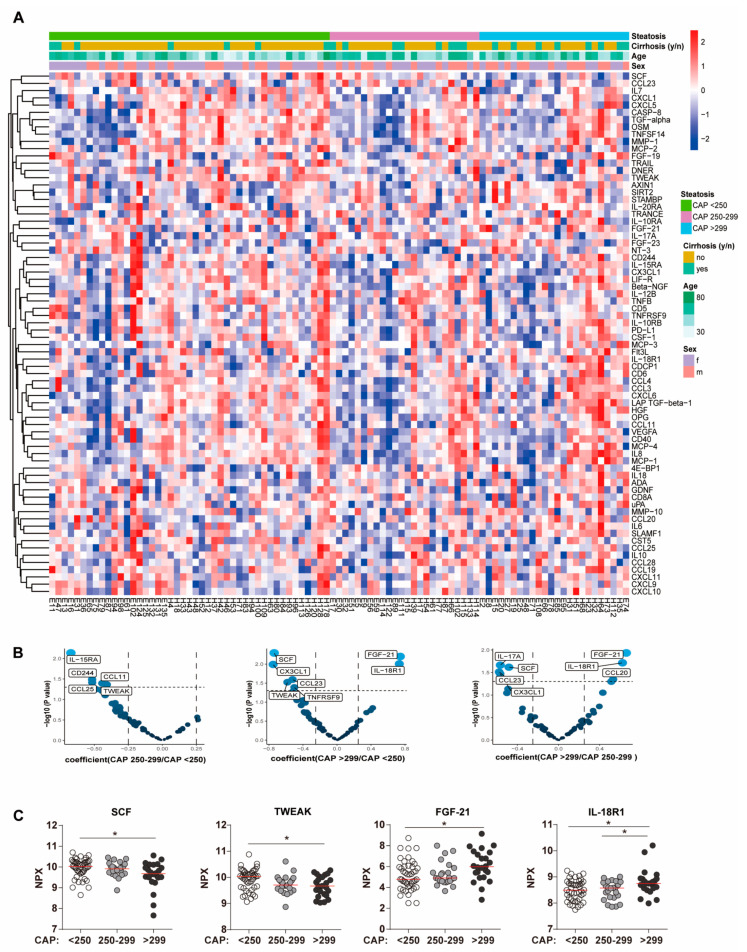
Four SIM markers differed in patients with differential status of steatosis after virus clearance. (**A**) Heatmap showing the expression pattern of 72 soluble inflammatory cytokines of each participant. Data are presented for 94 patients with chronic hepatitis C. (**B**) Volcano plots showing the difference between each two groups (CAP < 250 dB/m; CAP 250–299 dB/m; CAP > 299 dB/m) from the linear regression model after correction of age, gender and BMI. Size and color of the dot represent the −log10 (nominal *p* value), X axis indicates the coefficient. (**C**) Four markers that differed in patients with different grade of steatosis were summarized. Quantitative comparisons among multiple groups were performed by nonparametric Kruskal–Wallis test and Dunn’s multiple comparison test. * *p* < 0.05. Abbreviations: SCF, stem cell factor; TWEAK, tumor necrosis factor ligand superfamily member 12; FGF, fibroblast growth factor; and IL-18R1, interleukin-18 receptor 1.

**Figure 3 biomedicines-11-01529-f003:**
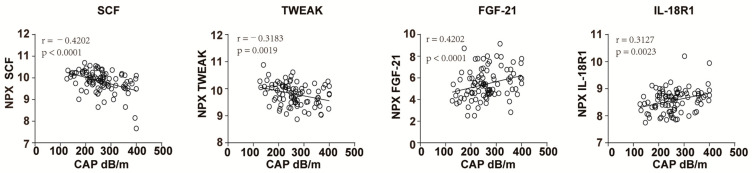
Correlation analysis of SCF, TWEAK, FGF-21, and IL-18R1 with CAP values. Linear regression of SCF, TWEAK, FGF-21, and IL-18R1 with CAP value in patients with chronic HCV infection at 96 weeks after the initiation of treatment. *p* value and R^2^ were calculated.

**Figure 4 biomedicines-11-01529-f004:**
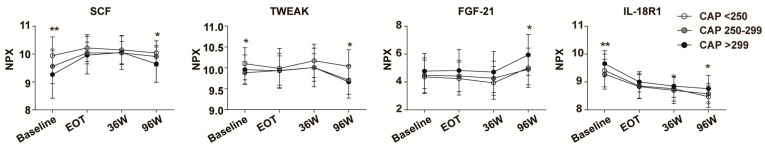
The expression of the four markers at four time points. The kinetics of the expression of the four markers at different time points were summarized. Statistical analysis was performed via ANOVA with Kruskal–Wallis test followed by Dunn’s multiple comparison test. * *p* < 0.05, ** *p* < 0.01. Median and standard errors of mean are presented. Abbreviations: SCF, stem cell factor; TWEAK, tumor necrosis factor ligand superfamily member 12; IL-18R1, interleukin-18 receptor 1.

**Table 1 biomedicines-11-01529-t001:** Characteristics of patients with chronic HCV infection at 96 w.

	CAP < 250	CAP 250–299	CAP > 299	*p* Value
CAP dB/m (96 w)	205 (100–249)	267.5 (250–293)	357 (302–400)	<0.0001
CAP dB/m (baseline)	220 (100–322)	238.5 (143–363)	307 (189–400)	<0.0001
Number	44	23	27	-
Age (y)	52 (28–83)	62 (27–71)	60 (37–74)	0.1700
Gender (male/female)	16/28	10/13	13/14	0.6044
Genotype	1B	1B	1B	-
BMI kg/m^2^ (96 w)	24.82 (19.00–33.33)	26 (20.60–39.62)	29.96 (22.59–45.67)	0.0033
BMI kg/m^2^ (baseline)	25.4 (19.00–32.97)	25.6 (19.92–37.76)	28.7 (21.80–43.20)	0.0002
Fibroscan kPa (96 w)	5.4 (2.4–22.2)	6.4 (3.1–42.3)	6.9 (4.2–60.9)	0.0005
Fibroscan kPa (baseline)	5.9 (3.1–47.2)	9.8 (3.1–45.7)	14 (4.4–60)	0.0092
With/without diabetes	6/38	6/17	7/20	0.3297
With/without alcohol consumption	2/42	2/21	3/24	0.5726

Note: unless otherwise indicated, values represent median (range). The statistical analysis of gender, diabetes, and alcohol consumption was performed via Fisher’s exact test. Kruskal–Wallis test was used for age, CAP, fibroscan and BMI. Abbreviations: CAP, controlled attenuation parameter; BMI, body mass index.

**Table 2 biomedicines-11-01529-t002:** Classification of SIMs based on their function.

Inflammation Panel	Pass the Quality Control in Total (*n* = 72)
Chemokines (*n* = 19)	MCP1, MCP2, MCP3, MCP4, CXCL1, CXCL5, CXCL6, CXCL9, CXCL10, CXCL11, CCL3, CCL4, CCL11, CCL19, CCL20, CCL23, CCL25, CCL28, CX3CL1
Interleukins (*n* = 12)	IL-6, IL-7, IL-8, IL-10, IL-10RA, IL-10RB, IL-12B, IL-15RA, IL-17A, IL-18, IL-18R1, IL-20RA
TNF-associated cytokines (*n* = 6)	TNFRSF9, TNFB, TNFSF14, TWEAK, TRANCE, TRAIL
Growth factor (*n* = 13)	Beta-NGF, GDNF, DNER, VEGFA, SCF, LAP, SCF-1, TGF, HGF, FGF-19, FGF-21, FGF-23, NT-3
Ligands or receptors (*n* = 9)	CDCP1, CD244, CD5, CD6, CD8A, CD40, PD-L1, LIF-R, Flt3L
Others (*n* = 13)	MMP-1, MMP-10, ADA, OPG, uPA, AXIN1, CST5, OSM, SLAMF, SIRT2, 4E-BP1, STAMBP, CASP-8

Abbreviation: ADA, adenosine deaminase; CASP-8, caspase 8; CCL, C-C motif chemokine; CDCP1, CUB domain-containing protein 1; CXCL, C-X-C motif chemokine; CST5, cystatin D; DNER, delta- and Notch-like epidermal growth factor-related receptor; 4E-BP1, eukaryotic translation initiation factor 4E-binding protein 1; FGF, fibroblast growth factor; GDNF, glial-cell-line-derived neurotrophic factor; HCF, hepatocyte growth factor; IL, interleukin; LAP, latency-associated peptide; LIF-R, leukemia inhibitory factor receptor; MCP, monocyte chemotactic protein; MMP, matrix metalloproteinase; NGF, nerve growth factor; OPG, osteoprotegerin; OSM, oncostatin-M; SCF, stem cell factor; SLAMF, signaling lymphocytic activation molecule; STAMBP, STAM-binding protein; TNF, tumor necrosis factor; TNFRSF, TNF receptor superfamily member; TNFSF, TNF superfamily member; TWEAK, tumor necrosis factor superfamily member 12; TRANCE, TNF-related activation-induced cytokine; TRAIL, TNF-related apoptosis-inducing ligand; TGF, transforming growth factor; VEGFA, vascular endothelial growth factor A.

## Data Availability

The data that support the findings of this study are available from the corresponding author upon reasonable request.

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
