# Peer review of "Controlled Attenuation Parameter Is Associated with a Distinct Systemic Inflammatory Milieu after Clearance of HCV Infection"

_biomedicines, 2023, doi:10.3390/biomedicines11061529_

Round 1
Reviewer 1 Report
1. The authors present an interesting work in which it is suggested that four biomarkers, SCF, TWEAK, FGF-21 and IL-18R1, might be specifically associated with hepatic steatosis after virus clearance. The methodology is adequate, and the techniques used in the work are very sensitive. At the end of the Discussion, the authors clearly highlight the strengths and limitations of the work to guide other researchers and readers.
2. The sentence that appears at the end of the Results: "Therefore, we speculated that these four markers may specifically be related to steatosis at the long-time follow-up after viral clearance", could be deleted because this same sentence is already mentioned in the Discussion section.
3. It is suggested that some comment be added in the Discussion of the results to suggest an explanation of why steatosis disorder remains in most of the patients despite successful viral clearance.
Some phrases should be revised or replaced
Reviewer 2 Report
I read with interest the paper entitled “Steatosis is associated with a distinct systemic inflammatory milieu after clearance of HCV infection”. The authors investigated the association of several inflammatory mediators with the steatosis grade 96 weeks after achieving SVR. The topic of the paper is interesting, original and currently popular, and the date presented might be important for better understanding of pathogenesis of HCV induced steatosis.
However, there are several issues that authors should address to improve the manuscript:
1.) The Study design should be more clearly written:
a) Why steatosis grade II/III was classified according to CAP >300db/m? This is not commonly used classification, and reference [18] is not correctly cited.
b) It is unclear if these patients have already been included in other publications? Please provide the timeframe of patients’ recruitment.
c) Why only genotype 1b patients were included? This should be mentioned in study limitations.
2.) Results section is hard to read:
a) Figure 1A and Table 1 should be part of Methods section
b) It seems more logical to move first paragraph of Results section (line 131-149) in subsection 3.2
c) Table 2 – provide additional data on patients’ baseline characteristics, specifically steatosis grade, CAP, Fibroscan and BMI before DAA treatment
d) Obesity is usually classified as BMI >30 not >28? Please provide median with IQR (or mean with SD) of BMI.
e) It would be interesting to show the data on AST, ALT, GGT… changes at week 0 and 96 (similarly as shown for CAP in Figure 1B)
f) Was the kinetics of all measured inflammatory markers analyzed or just 4 selected based on wk96 findings?
3.) Limitations should be more comprehensively written and include limitations mentioned above.
The paper requires moderate English editing.
Reviewer 3 Report
The study by Yanqin Du et al. is a prospective study evaluating the association between 92 soluble inflammatory mediators (SIMs) and the controlled attenuation parameter (CAP) value after sustained virological response (SVR) with direct-acting antivirals (DAA) for hepatitis C virus (HCV) in 94 patients. The authors classified the patients into 3 groups according to CAP values after SVR and found a good correlation with body mass index (BMI), liver stiffness measurement (LSM), ALT, GGT, and glucose. After evaluating the correlation between CAP and SIMs only 4 (SCT, TWEAK, FGF-21, and IL-18R1) showed moderate correlation (r=0,3-0,4). The authors concluded that these four biomarkers are associated with hepatic steatosis after SVR and that the potential role in the pathogenesis of post-SVR nonalcoholic steatohepatitis requires further investigation.
However, the current version has important limitations and authors should be much more prudent in interpreting their results to avoid wrong conclusions and to be accepted in biomedicines.
Major comments
1. Studies evaluating steatosis and liver fibrosis in liver biopsies after SVR are lacking. Changes in steatosis and the correlation between CAP and steatosis or LSM and fibrosis after SVR are not well-known (doi:10.1002/jmv.24950). Moreover, obesity (body mass index-BMI) is an important confounding factor when steatosis is evaluated by elastography using CAP (doi: 10.3390/diagnostics10110940). Therefore authors should change the word “steatosis” to “controlled attenuation parameter” in their study (title, abstract, results, and discussion section) to avoid wrong conclusions.
2. Introduction and Discussion.
· Please include references about BMI as a confounding factor of CAP and the lack of studies evaluating steatosis and fibrosis in liver biopsies after SVR.
3. Methods.
· Please, describe the type of elastography (transient elastography by FibroScan?) and the method to obtain the CAP values.
· Please include the quality criteria of CAP and references.
4. Results, Tables, and Figures.
· Please, categorize patients according to CAP values (<250, 250-299, >299) better than steatosis grade (S0, S1, or S2-3)
2. Results
· Please avoid speculations in the results section: Lines 158-160. Lines 203-205. Lines 231 and 233-34.
· Differential protein analysis by linear regression should be corrected not only by age and gender but also by BMI
· How could authors exclude that their results are related to steatosis and not associated with obesity or diabetes? Authors should show the results categorizing patients according to BMI, DM, and CAP. How were the CAP values in those with low BMI or without DM?
· Please, show the kinetics of these four biomarkers according to the changes in BMI during the time. Authors should show marker kinetics in patients with an increased or decreased BMI after SVR.
Round 2
Reviewer 2 Report
The authors have sufficiently responded to all my comments and significantly improved the manuscript.
I do not have any further complaints.
Minor editing of English language is required.
Author Response
The authors have sufficiently responded to all my comments and significantly improved the manuscript.
I do not have any further complaints.
Author's response
We thank the reviewer for previous important points that helped us improve the manuscript and thank the reviewer for this positive feedback.
Reviewer 3 Report
The study by Yanqin Du et al. has significantly improved after following the editor and reviewers’ recommendations. However, two important topics must be considered to avoid wrong conclusions.
First, the studies evaluating, in liver biopsies, the evolution of steatosis (PMID: 16856622) and liver fibrosis (PMID: 34525253) after SVR are very lacking. Therefore, changes in steatosis and fibrosis after SVR and the correlation with the controlled attenuation parameter (CAP) of FibroScan and liver stiffness measurements (LSM) are not well-known.
Second, previous studies have shown that SVR decreases LSM and CAP values rapidly independently of fibrosis or steatosis. Therefore the CAP and LSM cut-offs used to categorize steatosis and fibrosis in HCV-infected patients should not be used after SVR because their diagnostic accuracy could be different (PMID: 29494353, PMID: 34525253).
comments
1. Authors should change the word “steatosis” to “controlled attenuation parameter” in their study (title, abstract, results, and discussion section) to avoid wrong conclusions.
2. Please, categorize patients according to CAP values (<250, 250-299, >299) better than steatosis grade (S0, S1, or S2-3)
Author Response
The study by Yanqin Du et al. has significantly improved after following the editor and reviewers’ recommendations. However, two important topics must be considered to avoid wrong conclusions.
First, the studies evaluating, in liver biopsies, the evolution of steatosis (PMID: 16856622) and liver fibrosis (PMID: 34525253) after SVR are very lacking. Therefore, changes in steatosis and fibrosis after SVR and the correlation with the controlled attenuation parameter (CAP) of FibroScan and liver stiffness measurements (LSM) are not well-known.
Second, previous studies have shown that SVR decreases LSM and CAP values rapidly independently of fibrosis or steatosis. Therefore the CAP and LSM cut-offs used to categorize steatosis and fibrosis in HCV-infected patients should not be used after SVR because their diagnostic accuracy could be different (PMID: 29494353, PMID: 34525253).
Author’s response
We thank the reviewer for these important points that helped us improve the manuscript. We agree with the reviewer that it is not accurate the use CAP to categorize steatosis in HCV-infected patients after SVR.
comments
- Authors should change the word “steatosis” to “controlled attenuation parameter” in their study (title, abstract, results, and discussion section) to avoid wrong conclusions.
Author’s response
We thank the reviewer for this suggestion. Accordingly, we revised the word “steatosis” to “controlled attenuation parameter” in our study in title, abstraction, results and discussion section.
2. Please, categorize patients according to CAP values (<250, 250-299, >299) better than steatosis grade (S0, S1, or S2-3).
Author’s response
We thank the reviewer for this suggestion. Therefore, we categorized patients according to CAP values (<250, 250-299, >299), and revised all the figure, figure legends and results part accordingly.